# The Antioxidant *Dendrobium officinale* Polysaccharide Modulates Host Metabolism and Gut Microbiota to Alleviate High-Fat Diet-Induced Atherosclerosis in ApoE^−/−^ Mice

**DOI:** 10.3390/antiox13050599

**Published:** 2024-05-13

**Authors:** Jingyi Qi, Shuaishuai Zhou, Guisheng Wang, Rongrong Hua, Xiaoping Wang, Jian He, Zi Wang, Yinhua Zhu, Junjie Luo, Wenbiao Shi, Yongting Luo, Xiaoxia Chen

**Affiliations:** 1Key Laboratory of Precision Nutrition and Food Quality, Department of Nutrition and Health, China Agricultural University, Beijing 100193, China; zb20213311055@cau.edu.cn (J.Q.); shuaishuaizhou@cau.edu.cn (S.Z.); sy20223313571@cau.edu.cn (Z.W.); zhuyinhua@cau.edu.cn (Y.Z.); luojj@cau.edu.cn (J.L.); 2Department of Radiology, The Third Medical Centre, Chinese PLA General Hospital, Beijing 100039, China; wangguisheng@301hospital-3.mil.cn (G.W.); huarongrong@301hospital-3.mil.cn (R.H.); 3Zhejiang Medicine Co., Ltd., Shaoxing 312366, China; wangxiaoping@zmc.top; 4National Center of Technology Innovation for Dairy, Hohhot 010110, China; hejian@yili.com

**Keywords:** atherosclerosis, *Dendrobium officinale* polysaccharide, antioxidant, anti-inflammatory, gut microbiota

## Abstract

Background: The discovery of traditional plants’ medicinal and nutritional properties has opened up new avenues for developing pharmaceutical and dietary strategies to prevent atherosclerosis. However, the effect of the antioxidant *Dendrobium officinale* polysaccharide (DOP) on atherosclerosis is still not elucidated. Purpose: This study aims to investigate the inhibitory effect and the potential mechanism of DOP on high-fat diet-induced atherosclerosis in Apolipoprotein E knockout (ApoE^−/−^) mice. Study design and methods: The identification of DOP was measured by high-performance gel permeation chromatography (HPLC) and Fourier transform infrared spectroscopy (FTIR). We used high-fat diet (HFD)-induced atherosclerosis in ApoE^−/−^ mice as an animal model. In the DOP intervention stage, the DOP group was treated by gavage with 200 μL of 200 mg/kg DOP at regular times each day and continued for eight weeks. We detected changes in serum lipid profiles, inflammatory factors, anti-inflammatory factors, and antioxidant capacity to investigate the effect of the DOP on host metabolism. We also determined microbial composition using 16S rRNA gene sequencing to investigate whether the DOP could improve the structure of the gut microbiota in atherosclerotic mice. Results: DOP effectively inhibited histopathological deterioration in atherosclerotic mice and significantly reduced serum lipid levels, inflammatory factors, and malondialdehyde (F/B) production. Additionally, the levels of anti-inflammatory factors and the activity of antioxidant enzymes, including superoxide dismutase (SOD) and glutathione peroxidase (GSH-PX), were significantly increased after DOP intervention. Furthermore, we found that DOP restructures the gut microbiota composition by decreasing the *Firmicutes*/*Bacteroidota* (F/B) ratio. The Spearman’s correlation analysis indicated that serum lipid profiles, antioxidant activity, and pro-/anti-inflammatory factors were associated with *Firmicutes*, *Bacteroidota*, *Allobaculum*, and *Coriobacteriaceae_UCG-002*. Conclusions: This study suggests that DOP has the potential to be developed as a food prebiotic for the treatment of atherosclerosis in the future.

## 1. Introduction

Atherosclerosis is a fundamental trigger for ischemic stroke, myocardial infarction, and many peripheral arterial diseases [1]. Epidemiological studies suggest that the prevalence of atherosclerosis is increasing globally due to adopting a Western lifestyle [2]. In addition, hyperlipidemia, hypertension, diabetes mellitus, and age are also common triggers of atherosclerosis [3,4]. Therefore, gaining a deeper understanding of the pathological mechanisms of atherosclerosis helps improve its prevention and treatment.

Atherosclerosis is a complex pathological process involving inflammatory and oxidative responses in blood vessels [5]. The increased levels of low-density lipoprotein cholesterol and free radicals stimulate endothelial cells to release chemokines, which subsequently trigger the accumulation of monocytes and lymphocytes within the aortic wall [6]. Smooth muscle cells in the deep layers of the vessel wall migrate deeper into the vascular endothelium and form fibrous caps with lipids and collagen [7,8]. With the further deposition of lipids, the aggregation of phagocytes and the release of inflammatory factors are further aggravated [9,10]. The fibrous cap gradually becomes thinner and evolves into unstable plaques [11]. The treatment of atherosclerosis is mainly based on etiological treatment [12,13,14,15]. For example, statins interfere with atherosclerosis by lowering blood lipid levels [16,17]. However, patients taking statins can experience many side effects, such as liver damage, rhabdomyolysis, and hyperglycemia [18,19].

The efficacy of Chinese herbal medicine is relatively mild, usually does not cause strong stimulation to the body, and can also be used in many clinical practices [20]. For more than two thousand years, Chinese herbal medicine has been reported as an effective treatment for atherosclerotic diseases because of its lipid-lowering [21] and antioxidant or anti-hypertension activity [22]. Therefore, the identification of new Chinese herbal medicines and their bioactive components with efficacy and safety is a promising way to prevent and treat patients with atherosclerosis.

*Dendrobium officinale* has been widely studied for its commercial and medicinal value. *Dendrobium officinale* is a folk herbal medicine and contains many anti-oxidative compounds, such as alkaloids, flavonoids, and polysaccharides [23]. Polysaccharides are difficult to degrade by enzymes encoded in the human genome. However, *Bacteroidetes*, *Firmicutes*, a small amount of *Actinobacteria*, and *Proteobacteria* in the intestinal flora can secrete polysaccharide-degrading enzymes to degrade polysaccharides and promote the digestion and absorption of polysaccharides [24,25]. The non-digestible components and fermentation products of polysaccharides can further regulate the composition of intestinal flora, improve intestinal function, and maintain body health [26,27]. DOP is the main ingredient of *Dendrobium officinale*, and it has antioxidant, anti-inflammatory, and anti-cancer effects. In addition, DOP was reported to influence the composition and metabolism of the gut microbiota [28]. However, whether the DOP alleviates atherosclerosis through its bioactivities remains unclear.

In this study, we found that DOP alleviated atherosclerosis through its lipid-lowering, antioxidant, and anti-inflammatory activities. Moreover, DOP ameliorated atherosclerosis by improving the structure of the gut microbiota in atherosclerotic mice. 

## 2. Materials and Methods

### 2.1. Preparation and Identification of DOP

DOP (≥95% purity, cat: TDT013) [29] was obtained from Shanghai Ronghe Pharmaceutical Technology Development Company. The monosaccharide composition of the DOP was determined using high-performance liquid chromatography (HPLC, Shimadzu LC-20AD, Japan). In detail, 3 mL of trifluoroacetic acid (TFA, 2 M) was added to 2 mg of DOP and hydrolyzed in a sealed tube at 120 °C for 3 h. After cooling, the sample was mixed with methanol, and the evaporation was repeated three times to remove excess TFA. A total of 250 μL of 0.6 mol/L NaOH and 500 μL of 0.4 mol/L 1-phenyl-3-methyl-5-pyrazolone (PMP) methanol solution were added to the hydrolysate or standard monosaccharide mixture (1 mg/mL). The reaction was conducted at 70 °C for 1 h and cooled in cold water for 10 min. A total of 500 μL of 0.3 mol/L HCl was added to terminate the reaction, and after three extractions with chloroform, the supernatants obtained were used in HPLC. An Xtimate-C18 column (4.6 mm × 200 mm, 5 μm) was used to separate the sample at 30 °C. The mobile phase was composed of acetonitrile and 0.1 M phosphate buffer (pH = 6.7) at a ratio of 17:83 (*v*/*v*), with a flow rate of 1 mL/min. The same chromatographic conditions were used for the monosaccharide standards. The peak area of the standard solution was compared with that of the sample to calculate the content of monosaccharides in the DOP sample. The external standard method was used to calculate the molar ratio of monosaccharides.

The molecular weight of DOP was measured by high-performance gel permeation chromatography (HPGPC). A total of 1 mg DOP was ground with 100 mg dried KBr powder and then pressed into thin tablets. The DOP’s functional groups were scanned by an FTIR infrared spectrometer in the range of 4000–400 cm^−1^ (Nicolet6700 FTIR spectrometer, Thermo Fisher Scientific, Waltham, MA, USA).

### 2.2. Animal Experiments and Groups

Six-week-old male ApoE^−/−^ mice (22 ± 2 g) were obtained from Beijing Vital River Laboratory Animal Technology (Beijing, China). The mice were maintained under specific-pathogen-free conditions (temperature: 20–26 °C; humidity: 40–70%; pressure: 45 Pa; animal illumination: 15–20 Lux; light: 12 h/12 h light/dark cycle). After 1 week of acclimatization feeding, we induced a mouse model of atherosclerosis by feeding ApoE^−/−^ mice with a high-density lipoprotein diet (41% fat, 4.7 kcal g^−1^ energy) (cat: H10141, Beijing Henderson Biotechnology, Beijing, China) for ten weeks. After a successful 10-week high-fat diet-induced atherosclerosis model, mice were randomly divided into two groups: the control group (CTL group) and the DOP intervention group (DOP group), with 5 mice in each group. Each group of mice continued to be induced with a high-fat diet. However, the CTL group was treated by gavage with 200 μL of normal saline. DOP dose was estimated by a scaling factor according to the previous studies [30,31]. The DOP group was treated by gavage with 200 μL of a 200 mg/kg DOP deionized water solution. In the intervention stage, gavage was administered at regular times each day and continued for eight weeks. Weekly records of the body weight and the amount of feed consumed were kept during the intervention. After anesthesia, the mice were sacrificed. The obtained aortic tissues were stored in 4% formalin or at −80 °C for use in histopathological studies or serum biochemical experiments, respectively.

The animal study protocol was approved by the Committee on the Ethics of Animal Experiments of China Agricultural University (Approval Code: AW92503202-5-1; Approval Date: 29 May 2023).

### 2.3. Biochemical Assays

GLU assay kit (A154-1-1), GSP assay kit (A037-2-1), MDA assay kit (cat: A003-1), SOD assay kit (cat: A001-3), and GSH-PX assay kit (cat: A005-1) were used to measure the activities or levels of the MDA, SOD, GSH-PX in the serum, respectively, according to the manufacturer’s instructions (Nanjing Jiancheng Technology, Nanjing, China). A microplate reader (Thermo Fisher Scientific, Waltham, MA, USA) was used to measure the absorbance at 505, 530, 532, 450, and 412 nm, respectively.

### 2.4. Analysis of Serum Lipid Profiles

Blood samples were collected and left for 2–3 h at room temperature. The serum was centrifuged at 3000 rpm for 15 min, divided, and stored at −80 °C. TC, TG, LDL-C, and HDL-C levels in serum were determined using commercially available assay kits (cat: F002-1-1, F001-1-1, A113-2-1, and A112-2-1, Nanjing Jiancheng Bioengineering Institute, China).

### 2.5. Real-Time qPCR Analysis

Total RNA was extracted using the TRIZOL reagent (cat: CW0580S, Invitrogen, Carlsbad, CA, USA). Aortas were homogenized with TRIZOL reagent, vortexed for 1 min with chloroform, and centrifuged at 1.2 × 10^4^ rpm for 15 min at 4 °C, thus generating two phases. The upper aqueous phase (containing RNA) was precipitated with isopropanol at −20 °C for 30 min and centrifuged at 1.2 × 10^4^ rpm for 10 min. RNA pellets were washed with 75% (*v*/*v*) ethanol, air-dried, and dissolved. Then, total RNA was reverse transcribed into first-strand cDNA with oligo (dT) primers using reverse transcriptase (cat: R323-01, Vazyme, Nanjing, China). Subsequently, the first-strand cDNA synthesis reaction mixture was used for PCR amplification (cat: Q712-02, Nanjing Vazyme Biotech, Nanjing, China). All primers used are shown in Appendix A.

### 2.6. Oil Red O Staining

Oil Red O staining is widely used to measure lipid levels of atherosclerosis in aorta tissues [32,33]. The sections were stained with a modified Oil Red O solution using a commercial assay kit (cat: G1261, Beijing Solarbio Science and Technology, Beijing, China). The stained slices were examined using a Leica CTR6 microscope (Leica, Wetzlar, Germany). The Oil Red O content was analyzed and quantified by the ImageJ software (version 1.8.0).

### 2.7. Hematoxylin–Eosin (H&E) Staining

Aortic tissue was fixed with 4% paraformaldehyde overnight and embedded in paraffin. The paraffin blocks were cut into thin slices (5 μm thickness) for further H&E staining. The H&E stain slices were captured by the optical microscope (Leica CTR6, Leica, Germany). ImageJ software (version 1.8.0) was used to analyze and quantify the atherosclerosis plaques.

### 2.8. Masson’s Trichrome Stain

Masson’s trichrome stain kit (G1340, Beijing Solarbio Science and Technology, Beijing, China) was used to stain the prepared paraffin sections (G1340, Beijing Solarbio Science and Technology, Beijing, China). The Masson’s trichrome stain slices were observed by the optical microscope (Leica CTR6, Leica, Germany). ImageJ software (version 1.8.0) was used to quantify the collagen proportion of stained pieces.

### 2.9. Immunofluorescence Analyses

For immunofluorescence, one set of aortic tissue sections was immunostained with primary antibodies against α-SMA (1:100, cat: 48938, Cell Signaling Technology, Danvers, MA, USA) in combination with secondary antibody conjugated with Alexa Fluor 555 (goat anti-mice, 1:300); the other set of sections was immunostained with primary antibodies against Mac-3 (1:100, cat: 108501, Biolegend, San Diego, CA, USA) in combination with secondary antibody conjugated with Alexa Fluor (goat anti-Mouse, 1:300). The sections were visualized using a confocal laser scanning microscope (Zeiss, LSM 780, Oberkochen, Germany).

### 2.10. Gut Microbiota Analysis

The genomic DNA was extracted from different samples using the CTAB/SDS method [34]. The QIIME (Quantitative Insights Into Microbial Ecology) software package (version 1.2.8) was used for diversity analyses [35], and the high-quality reads were collected into the operational classification unit (OTU) for a series of analyses [36]. The PICRUSt pipeline was used to predict the abundance of Kyoto Encyclopedia of Genes and Genomes (KEGG) orthologs [37], and the results were analyzed for group differences using Statistical Analysis of Taxonomic and Functional Profiles (STAMP) [38].

### 2.11. Statistical Analysis

The data were expressed as mean ± standard deviation (SD) in this study. Significant differences between the DOP group and the CTL group were analyzed using the one-way analysis of variance (ANOVA) method. All calculations were performed using GraphPad Prism software V. 9.0 (GraphPad Software, San Diego, CA, USA). In all assays, *p* < 0.05 was presented as statistically significant.

## 3. Results

### 3.1. Structural Analysis of DOP

The heavy mean molecular weight (Mw) and the number mean molecular weight (Mn) of DOP were calculated to be 4.5 kDa and 1.1 kDa, respectively, with a polydispersity (Mw/Mn) of 4.20 (Appendix A). The monosaccharide composition analysis of DOP showed that the main components of DOP were mannose (Man) and glucose (Glc), with a molar ratio of 1.16:1.0 (Figure 1A,B). The FTIR spectra detected from 400 to 4000 cm^−1^ of DOP displayed characteristic bands at 603, 1077, 1253, 1404, 1612, 2927, and 3415 cm^−1^ (Figure 1C). The broad absorption peaks at 3415 cm^−1^ correspond to O–H stretching, and the band at 2927 cm^−1^ belongs to the –C–H stretching vibration. In addition, the peaks at 1612 cm^−1^ may be the C=O stretching vibration. The band at 1404 cm^−1^ was attributed to –C–H stretching, while the bands at 1253 cm^−1^ and 1077 cm^−1^ correspond to –O–H and –C–O stretching. And the band at 603 cm^−1^ may be the bending vibration of –C–CO–C– [39].

### 3.2. Effects of DOP on Serum Lipid Levels in HFD-Induced Atherosclerosis Mice

ApoE^−/−^ mice have been frequently used in atherosclerosis research [40,41]. The model of HFD-induced atherosclerosis has been commonly used in previous studies [42,43]. Herein, we fed ApoE^−/−^ mice with an HFD for 18 weeks and simultaneously treated them with DOP or control saline for 8 weeks. (Appendix A). No overt differences were observed between the DOP group and the CTL group concerning body weight and food intake (Appendix A). Hyperglycemia and hyperlipidemia are important indicators to evaluate the risk of atherosclerosis [44]. We first investigated the levels of glucose (GLU) and glycosylated serum protein (GSP) after DOP treatment. The results showed that the oral administration of DOP had no significant hypoglycemic effect (*p* > 0.05) (Appendix A). We next investigated whether DOP treatment decreased serum lipid profile levels in the HFD-induced atherosclerosis mice. Interestingly, the serum lipid profile in the DOP-treated mice changed compared with the control saline-treated mice. Serum TC, TG levels, and LDL-C levels are significantly decreased (*p* < 0.01). In contrast, serum HDL-C levels are increased in the DOP-treated group (*p* < 0.001) (Figure 2A), which indicates that DOP intervention significantly impacts the serum lipid profiles in atherosclerosis mice.

### 3.3. Effects of DOP on the Antioxidant Capacity and Inflammation of Atherosclerosis Mice

An increasing number of studies indicate that excessive oxidative stress can promote the occurrence and development of atherosclerosis [45,46]. To investigate whether DOP treatment improved antioxidant activities in HFD-induced atherosclerosis mice, we measured oxidative stress indicators (GSH-PX, MDA, and SOD) in the serum. GSH-PX activity in serum was significantly increased in the DOP group (*p* < 0.001) compared with the control group (Figure 2B). SOD was also significantly increased in serum in the DOP group compared to the control group (*p* < 0.001) (Figure 2C). The decrease in MDA levels reflected the improvement in cell membrane lipid peroxidation (*p* < 0.01) (Figure 2D). These results suggested that DOP effectively inhibited the development of atherosclerosis by reducing oxidative damage.

Inflammation is another important clinical manifestation of atherosclerosis [47,48]. Here, mRNA levels of pro-inflammatory and anti-inflammatory factors in aortic tissues were measured. Our results showed that DOP supplementation significantly reduces inflammation factor levels (*Tnf-α*, *Il-1β*, and *Il-6*) in HFD-fed mice (*p* < 0.01) (Figure 2E). Moreover, DOP supplementation enhances the expression level of anti-inflammatory factors (*Arg1*, *Mrc1*, *Retnla*, and *Irf4*) (*p* < 0.01) (Figure 2E). These results suggest that DOP effectively ameliorates atherosclerotic oxidative stress and inflammatory responses in HFD-fed mice.

### 3.4. DOP Ameliorates Atherosclerosis

The metabolism disorder of atherosclerosis will promote the progress of atherosclerosis evolution, specifically reflected in the enlargement of plaque, the growth of lipid deposition, the necrotic core, the increase in collagen content, and the deterioration of plaque stability [47,48,49,50]. To examine the DOP’s effects on atherosclerosis, we performed a histological analysis of consecutive sections to quantify the development of atherosclerotic lesions. Compared with the control saline-treated mice, the lipid deposition area and atherosclerotic plaque size in the DOP-treated mice were significantly reduced (*p* < 0.001) (Figure 3A–C). HE Staining experiments revealed that the control group plaques had undergone more severe plaque progression (Figure 3D). In contrast, DOP-treated mice had much less advanced plaques (Figure 3E). The lesion area of atherosclerotic plaques was reduced by 46% after DOP treatment compared with the CTL group (*p* < 0.001) (Figure 3F). Moreover, we observed smaller necrotic cores in the plaques of the DOP-treated mice (*p* < 0.001) (Figure 3G). The lesion area of necrotic cores in the plaques was reduced by 70% after DOP treatment compared with the CTL group (*p* < 0.001) (Figure 3H).

In addition, collagen content in the plaques was significantly higher in the DOP-treated mice (*p* < 0.001) (Figure 4A,B). Immunofluorescence staining showed greatly diminished Mac-3 (*p* < 0.001) (Figure 4C–E) and increased α-SMA expression (*p* < 0.001) (Figure 4F–H), suggesting that DOP enhances plaque stability. Collectively, these data demonstrate that DOP ameliorates HFD-induced atherosclerosis.

### 3.5. Effects of DOP on the Abundance and Diversity of Mouse Gut Microbiota

There is substantial evidence that the diverse microbial communities that colonize the human gastrointestinal tract profoundly affect health [51,52]. Therefore, we tested the impact of DOP intervention on fecal microbiota composition. Bacterial communities in fecal samples were characterized by 16S rRNA gene sequencing. Alpha diversity is often used to reflect the abundance and evenness of species in a community ecosystem. Interestingly, after an eight-week feeding of DOP, observed species, Chao1, Shannon, and Simpson indices were not significantly changed in the DOP group mice compared to the control group mice (Figure 5A–D), indicating that DOP supplementation does not alter the diversity of gut microbiota microorganisms in atherosclerosis mice. Similarities in the structure of the entire microbial community were evaluated using principal co-ordinates analysis (PCoA), with the two main principal components accounting for 41.7% and 22.2% of the total variance (*p* < 0.01) (Figure 5E). In addition, the four most abundant bacterial phyla in the feces of both DOP-treated and control mice were *Firmicutes, Bacteroidetes, Desulfobacterota,* and *Actinobacteria* (Figure 5F).

### 3.6. Differences in the Composition of Mice Gut Microbiota Structure by DOP Treatment

To investigate the role of gut microbiota in the progression of atherosclerosis, we analyzed the gut microbiota structure of mice at the phylum level. The results showed that the relative abundance of *Firmicutes*, *Bacteroidota*, and *Desulfobacterota* in the DOP group was significantly increased (*p* < 0.05). In contrast, the relative abundance of *Firmicutes* was significantly lower (*p* < 0.05) (Figure 5G,H). Compared to the F/B ratio observed in the control group (6.33), the F/B ratio exhibited a significant decrease (1.93) when DOP was administered.

To further investigate the gut microbiota differences between the DOP-treated mice and control mice, we analyzed the relative abundance of fecal samples at the genus level. Results indicated that *Allobaculum* and *norank_f__Muribaculaceae Ileibacterium* were the dominant genera in the intestinal tract of mice in the different treatments (Figure 5I,J). The relative abundance of *Allobaculum* in the DOP group was significantly increased compared to the control group (*p* < 0.05). The relative abundance of *unclassified_f__Lachnospiraceae* in the DOP group was significantly decreased (*p* < 0.05), and the relative abundance of *Coriobacteriaceae_UCG-002* was significantly increased (*p* < 0.05) compared to the control group. Thus, it is speculated that DOP may improve the symptoms of gut microbiota disorders in atherosclerosis mice by regulating the gut microbiota.

### 3.7. Correlation Analysis between Gut Microbiota and Metabolic Phenotypes

Spearman correlation analysis was performed to determine whether DOP-induced changes in metabolic phenotypes were associated with effects on the gut microbiota. The results showed that TC, TG, and LDL-C at the phylum level were positively correlated with *Firmicutes* and negatively correlated with *Bacteroidota* and *Proteobacteria* (*p* < 0.05). HDL-C, SOD, and GSH were negatively correlated with *Firmicutes* (*p* < 0.05), while positively correlated with *Bacteroidota* (*p* < 0.05). *Mrc1*, *Retnla*, and *Irf4* expression levels were strongly negatively correlated with *Firmicutes* and positively corrected with *Bacteroidota* and *Proteobacteria* (*p* < 0.05) (Figure 6A). At the genus level, TC, TG, and LDL-C were negatively correlated with *Allobaculum* and *Coriobacteriaceae_UCG-002* and positively correlated with *Blautia* (*p* < 0.05). HDL-C, SOD, and GSH were positively correlated with *Allobaculum* and *Coriobacteriaceae_UCG-002* and negatively correlated with Blautia (*p* < 0.05) (Figure 6B).

In addition, at the phylum level, there was a significant negative correlation between inflammatory factors (*Tnf-α*, *Il-1β*) and *Firmicutes* (*p* < 0.01) and a significant negative correlation between inflammatory factors (*Tnf-α*, *Il-1β*) and *Bacteroidota* and Proteobacteria (*p* < 0.05). Anti-inflammatory factors (*Mrc1*, *Retnla*, and *Irf4*) were negatively correlated with *Firmicutes* (*p* < 0.05) and were positively correlated with Bacteria (*p* < 0.01) and Proteobacteria (*p* < 0.05) (Figure 6A). At the genus level, there was a significant negative correlation between inflammatory factors (*Tnf-α*, *Il-1β*) and *Allobaculum* and *Coriobacteriaceae_UCG-002* (*p* < 0.05), and there was a significant positive correlation between inflammatory factors (*Tnf-α*, *Il-1β*) and *Blautia* (*p* < 0.05). Anti-inflammatory factors (*Arg1*, *Mrc1*, *Retnla*, and *Irf4*) were significantly negatively correlated with *Allobaculum* (*p* < 0.05) and *Coriobacteriaceae_UCG-002* (*p* < 0.05) (Figure 6B).

### 3.8. Effects of DOP on Microbial Community Functions Predicted by PICRUSt

In this study, the corresponding ID was obtained after standardizing the OTU abundance of each sample, and the abundance of KEGG available categories was calculated based on the OTU abundance. This study annotated gut microbial functions based on the KEGG database using PICRUS software (v2.3.0-b) to learn more about the functions with significant differences among different treatment groups. A comparison with the DOP group showed that six pathways were altered, mainly including aging (*p* < 0.01), development and regulation (*p* <0.05), immune system (*p* < 0.05), lipid metabolism (*p* < 0.05), immune diseases (*p* < 0.05), and cardiovascular disease (*p* < 0.001) (Figure 7). Therefore, DOP supplementation changes gut microbiota function in HFD-induced atherosclerosis mice.

## 4. Discussion

Overall, we compared the pathological phenotype, serum lipid levels, antioxidant activities, anti-inflammatory levels, and fecal microbiota composition of atherosclerotic mice that were treated with saline or DOP. The results showed that DOP treatment alleviated the development of atherosclerosis, reduced serum lipid levels and oxidative stress in atherosclerotic mice, and promoted the polarization of macrophages from the M1 phenotype to the M2 phenotype. Moreover, DOP alleviated atherosclerosis by modulating the gut microbiota, which contributed to the hypolipidemic, antioxidant, and anti-inflammatory activities.

Atherosclerotic plaques consist of extracellular lipid particles, foam cells (macrophages or smooth muscle cells that have phagocytosed large amounts of fat), and debris, which accumulate within the intima of the arterial wall and form the lipid deposition or necrotic core [53]. Our study showed that DOP treatment reduced atherosclerotic plaque burden and necrotic cores and slowed the progression of atherosclerosis. The necrotic core was encapsulated by a layer of collagen-rich matrix and endothelium-covered smooth muscle cells called the fibrous cap [7]. When the fibrous cap covering the necrotic core cracks, the plaque ruptures, exposing the core to blood flow. Researchers believe that vulnerable plaques are the underlying cause of most acute coronary events [54]. The two histological features that are more common in vulnerable plaques compared to stable plaques are a thinner fibrous cap and more inflammatory cell infiltration, respectively. Therefore, we analyzed the stability of atherosclerotic plaques using antibodies against Mac-3 (a macrophage marker) and α-SMA (a marker for smooth muscle cells) [55,56]. Our results showed that DOP-treated plaques had fewer macrophages and a thicker fibrous cap, suggesting that DOP treatment enhances plaque stability and reduces the risk of thrombus formation.

Atherosclerosis is a chronic disease with high mortality and an essential source of cardiovascular and cerebrovascular events. Hyperglycemia and hyperlipidemia are important indicators to evaluate the risk of atherosclerosis [44]. Previous studies have pointed out the anti-insulin resistance effects of DOP in different animal models, including leptin-deficient obese *ob*/*ob* mice, streptozotocin (STZ)-induced type 2 diabetes (T2D) mice, and diabetic nephropathy [57,58,59,60]. Therefore, DOP alleviates atherosclerosis, possibly partially by attenuating insulin resistance. However, lipid metabolism disorders are the primary cause of atherosclerosis. Improving lipid metabolism disorders and controlling blood lipids at appropriate levels have been shown to slow or even reverse the progression of atherosclerotic plaques [61,62]. It is well known that lipid metabolism disorders are manifested by high levels of TC, TG, and LDL-C in the blood or low levels of HDL-C [63]. In this study, we found that, compared with the control group, mice given 200 mg/kg DOP showed lower serum levels of TC, TG, and LDL-C and higher levels of HDL-C.

Animal studies have provided compelling evidence demonstrating the role of oxidative stress in atherosclerosis [45]. MDA is one of the end products of membrane lipid peroxidation. Its level is an important parameter reflecting the potential antioxidant capacity of the organism, which can reflect the rate and intensity of lipid peroxidation in the organism and also indirectly reflect the degree of peroxidative damage [64]. Antioxidant enzymes are essential antioxidant defense systems in vivo, including SOD and GSH-PX, which modify oxidative stress and thus alleviate atherosclerosis [65]. Our findings are in accord with recent studies indicating that DOP effectively protects the aorta from oxidant stress by increasing SOD and GSH-PX activity and inhibiting MDA production, indicating that DOP plays an antioxidant role by regulating the antioxidant oxidase system [66]. The bioactivity of polysaccharides is closely related to their structural properties, such as molecular weight and monosaccharide composition [67]. In this study, the molecular weight (Mw) of DOP is 4.5 kDa, and the main components of DOP were mannose (Man) and glucose (Glc), with a molar ratio of 1.16:1.0. Previous studies found that the smaller the molecular weight and the higher the mannose content of DOP, the stronger the antioxidant activity [68,69]. Therefore, the antioxidant activity of DOP in our study is stronger compared with the high Mw and lower mannose content of DOP in a previous study [57].

Inflammation runs through the formation and development of atherosclerosis [47]. Macrophages are the primary immune cell population in atherosclerotic lesions and have an essential role in the development of atherosclerosis [10]. There are two distinct macrophage subtypes in atherosclerotic plaques, namely, M1 phenotype macrophages and M2 phenotype macrophages [70]. M1 phenotype macrophages increase or maintain inflammatory responses by producing pro-inflammatory cytokines, and M2 phenotype macrophages are anti-inflammatory macrophages that are involved in tissue repair. The balance between the M1 phenotype macrophage and the M2 phenotype macrophage is involved in the pathogenesis of atherosclerosis. In our study, the levels of M1 phenotype macrophage biomarkers (pro-inflammatory factor) were reduced. On the contrary, the M2 phenotype macrophage biomarker (anti-inflammatory factors) levels were increased, suggesting that macrophages in atherosclerosis are polarized from the M1 phenotype to the M2 phenotype after DOP treatment. Compared with the M1 phenotype macrophage, the increased M2 phenotype macrophage could phagocytose apoptotic foam cells in atherosclerotic plaques, thus promoting tissue repair and inhibiting the progression of atherosclerosis.

The gut microbiota profoundly impacts the gut environment, affecting the function and metabolism of peripheral tissues [52]. Homeostasis of the gut microbiota is critical to host health. Previous studies have demonstrated that DOP is indigestible and non-absorbing after oral administration [71]. However, intestinal flora could degrade the DOP into short-chain fatty acids (SCFAs), exhibiting the potential of prebiotics and maintaining intestinal homeostasis [72]. In this study, DOP intervention regulates the entire microbial community structure. The ratio between *Firmicutes* and *Bacteroidetes* (F/B) is a typical marker of gut dysbiosis and implicates a predisposition to disease states [73,74]. Our study showed that DOP supplementation could increase *Bacteroidetes* and decrease *Firmicutes*. The ratio of F/B was better in the DOP groups than in the control mice. This indicates that DOP has a greater tendency to inhibit *Firmicutes*, thus tipping the balance in favor of *Bacteroidetes* proliferation in the gut, which could positively regulate the intestinal microecology and help prevent obesity. Therefore, the decrease in lipid levels in the DOP group compared to the control group after eight weeks of gavage may also be related to restoring homeostasis in the gut microbiota. Correlation analysis also indicates that the increase in abundance of *Coriobacteriaceae_UCG-002* is negatively associated with TC, TG, and LDL-C. *Coriobacteriaceae_UCG-002* has been reported to belong to the Actinobacteria species [75,76], but it has been found to play a role in reducing the risk of obesity in the presence of a high-fat diet [77]. In addition, KEGG pathway enrichment analysis showed that the differential flora was closely related to the lipid metabolism pathway. We suppose that DOP might reduce serum lipid levels by up-regulating lipid metabolism pathways in atherosclerotic mice, and more molecular biology experiments are needed to explore the mechanisms.

In this study, we found that DOP could increase the abundance of *Allobaculum* in atherosclerosis mice. *Allobaculum* is a crucial bacteria in the human gut that produces short-chain fatty acids [78]. As the final metabolites of intestinal microorganisms, short-chain fatty acids decrease the pH level in the gut, prevent the growth of harmful bacteria, and encourage the proliferation of beneficial bacteria, thus maintaining a balanced microecological environment in the intestine [79]. These fatty acids provide energy for intestinal mucosal cells, promote the growth of intestinal epithelial cells, and help preserve the intestinal barrier and resist inflammation [80]. Therefore, the *Allobaculum* plays a critical role in maintaining the integrity of the intestinal wall and constructing the intestinal immune system. The increase in the *Allobaculum* significantly inhibits the progression of atherosclerosis. In addition, our KEGG pathway enrichment analysis showed that the differential flora was closely related to the immune system and immune disease pathways, indicating that DOP plays a vital role in regulating macrophage polarization and increasing the abundance of *Allobaculum*, thereby contributing to the immune system’s normal functioning.

There are some limitations to be noted in our study. Some parameters, such as TC, TG, fatty acid composition, or SCFAs in feces, would be relevant to further exploring the potential mechanism of DOP in treating atherosclerosis by modulating the gut microbiota. Overall, DOP may affect host metabolism by altering the composition of the gut microbiota (Figure 8). DOP has been shown to reduce serum lipid levels, aortic inflammation, and oxidative damage. Additionally, DOP helps maintain a healthy gut microbiota and promotes a stable intestinal environment. These results suggest that DOP has a brilliant prospect for the development of food prebiotics or clinical adjuvants for the treatment of atherosclerosis.

## Figures and Tables

**Figure 1 antioxidants-13-00599-f001:**
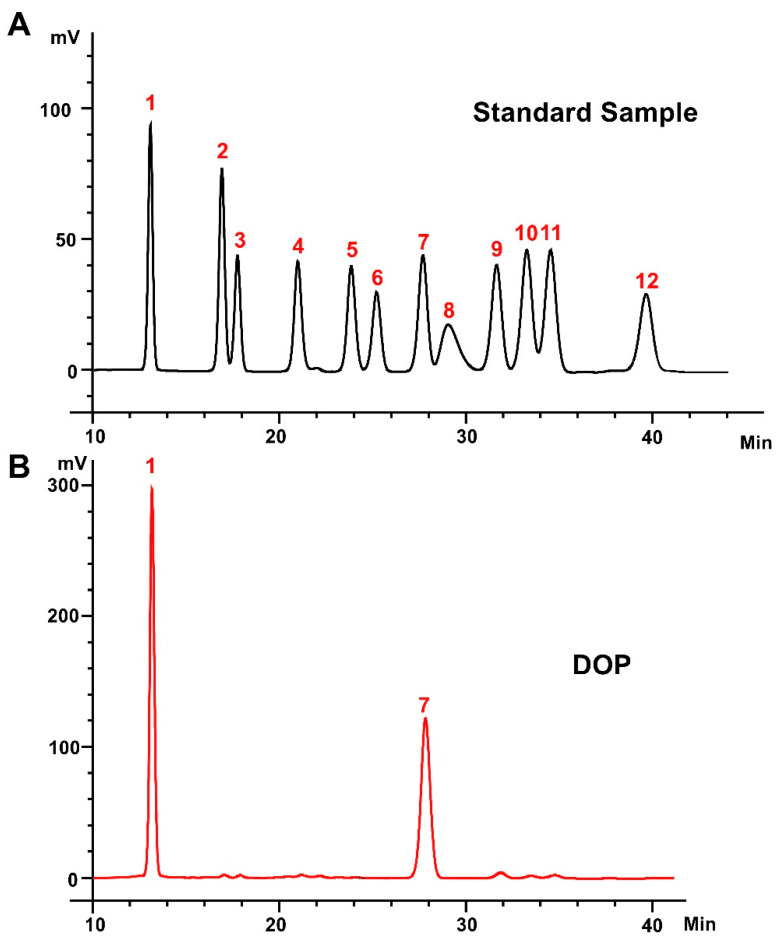
**Monosaccharide composition and FTIR analysis.** (**A**) Standard monosaccharides; (**B**) the composition of DOP by HPLC analysis: 1: mannose; 2: ribose; 3: rhamnose; 4: glucosamine; 5: glucuronic acid; 6: galacturonic acid; 7: galactosamine; 8: glucose; 9: galactose; 10: xylose; 11: arabinose; 12: fucose. (**C**) FTIR spectrum of DOP.

**Figure 2 antioxidants-13-00599-f002:**
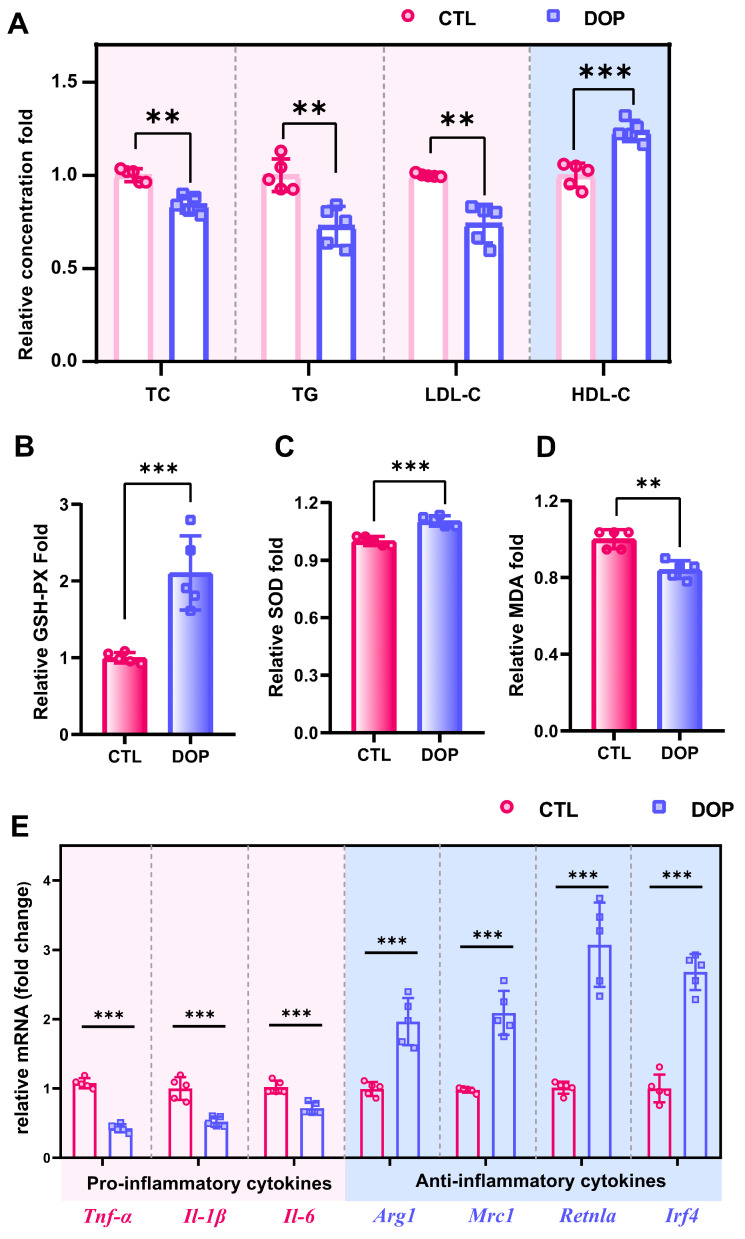
**DOP improves host metabolism in atherosclerosis-infected mice.** (**A**) The serum levels of TC, TG, LDL-C, and HDL-C after treatment with DOP. Effects of DOP intervention on GSH-PX (**B**), SOD (**C**), and MDA (**D**) in serum. (**E**) The mRNA levels of genes related to inflammatory (*Tnf-α*, *Il-1β*, and *Il-6*) and anti-inflammatory markers (*Arg1*, *Mrc1*, *Retnla*, and *Irf4*) in aorta tissues. All data are presented as mean ± SD. Statistical analysis was performed with a one-way ANOVA. ** *p* < 0.01, *** *p* < 0.001, compared with the control group.

**Figure 3 antioxidants-13-00599-f003:**
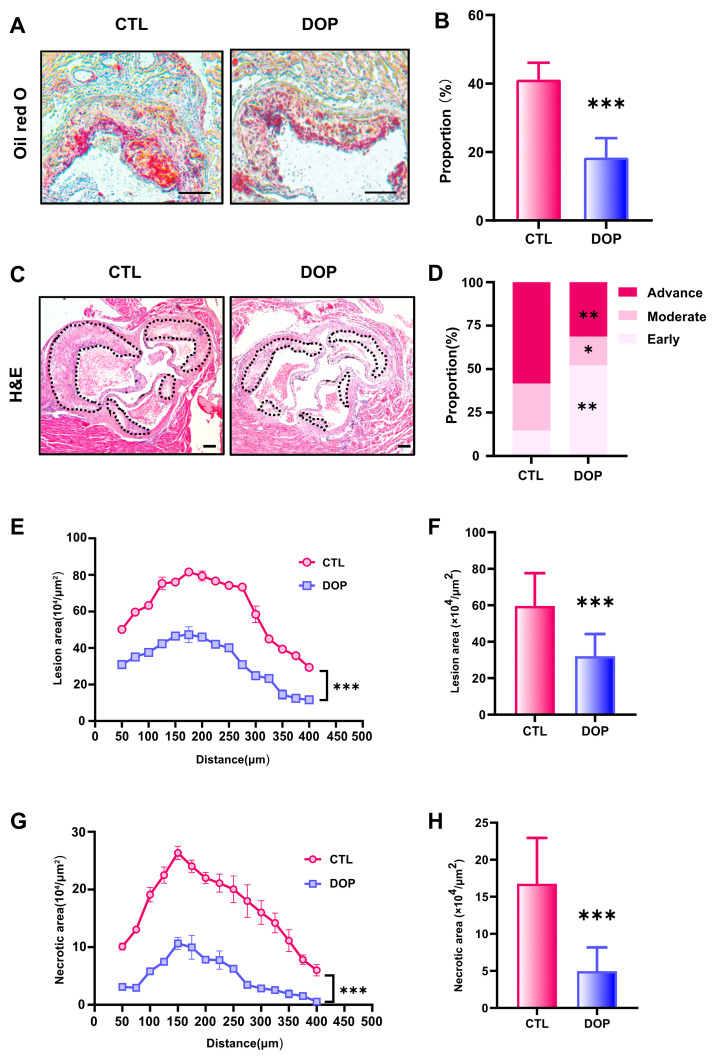
**DOP supplement leads to lower atherosclerosis burden in HFD-fed ApoE^−/−^mice.** (**A**) Representative images of Oil Red O staining of lesions isolated from the control group and DOP group. Scale bars = 100 μm. (**B**) The quantification of Oil Red O-positive plaque area (n = 5, at least 10 sections per mouse). (**C**) Representative H&E staining of the aorta tissues. Scale bars = 100 μm. (**D**) The distribution of early, moderate, and advanced plaques is based on the histological staging of the atherosclerotic lesions. (**E**) Lesion area of plaques on the aortic roots of each group of mice, presented for each group across the 400 µm of the aortic root (n = 5). (**F**) Quantification of the lesion area of aortic plaques (n = 5, at least 10 sections per mouse). (**G**) Curve chart of atherosclerotic necrotic core area (n = 5, at least 10 sections per mouse). (**H**) Quantification of the atherosclerotic necrotic core area of aortic plaques (n = 5, at least 10 sections per mouse). All data are presented as mean ± SD. Statistical analysis was performed with a one-way ANOVA. * *p* < 0.05, ** *p* < 0.01, *** *p* < 0.001, compared with the control group.

**Figure 4 antioxidants-13-00599-f004:**
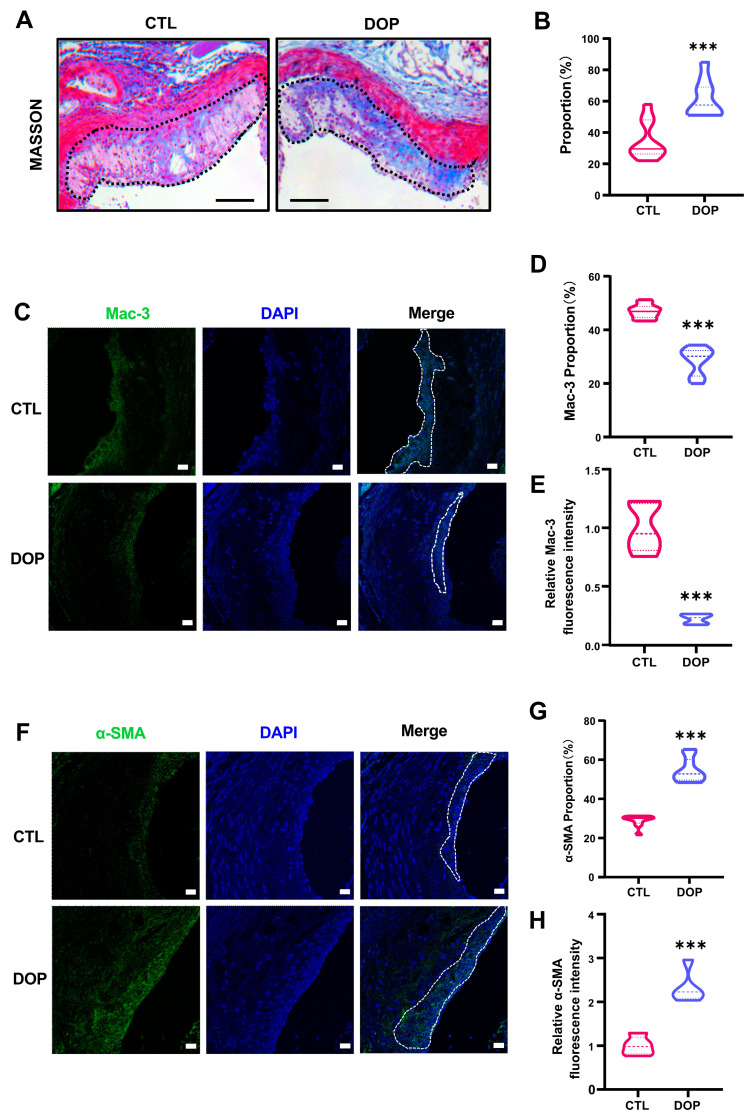
**The DOP supplement increases aorta stability in HFD-induced atherosclerosis mice.** (**A**) Representative Masson Trichrome (collagen) staining of aortic plaques Scale bars = 200 μm. (**B**) Quantification of Masson’s trichrome (collagen) staining of aortic plaques. (n = 5, at least 10 sections per mouse). (**C**) Staining of Mac3-positive macrophages (green) in the lesion area of the aortic root from the control group and DOP group, with representative images (**D**,**E**) of quantitative data of Mac3-positive macrophages in the lesion area and relative Mac-3 fluorescence intensity (n = 5 mice per group, at least 10 sections per mouse). Scale bars = 20 μm. (**F**) Staining of α-SMA-positive macrophages (green) in the lesion area of the aortic root from the control group and DOP group, with representative images (**G**,**H**) quantitative data of α-SMA-positive macrophages in the lesion area and relative α-SMA fluorescence intensity (n = 5, at least 10 sections per mouse). Scale bars = 20 μm. All data are presented as mean ± SD. Statistical analysis was performed with a one-way ANOVA. *** *p* < 0.001, compared with the control group.

**Figure 5 antioxidants-13-00599-f005:**
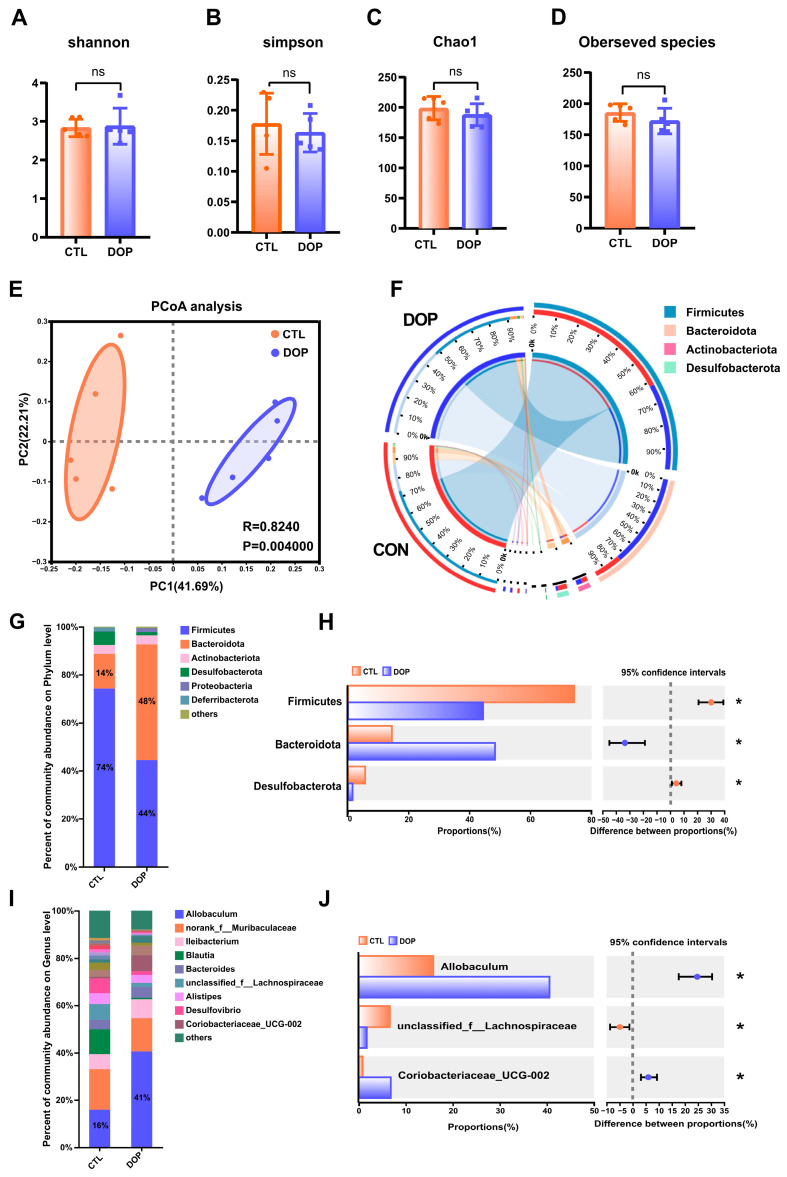
**Effects of DOP intervention on the structure and function of fecal microbial community.** (**A**–**D**) α-diversity, including the bacterial diversity index (Shannon and Simpson) and the richness index (Chao1 and observed species). (**E**) PCoA analysis of the intestinal flora of mice in the two groups. (**F**) Phylum of intestinal flora level. (**G**) The difference in composition of the intestinal flora of mice in the two groups is on the phylum level. (**H**) The difference in composition of control and DOP at the phylum level. (**I**) The difference in composition of intestinal flora at the genus level in three groups of mice. (**J**) The difference in composition of control and DOP at the genus level. All data are presented as mean ± SD. Statistical analysis was performed with a one-way ANOVA. * *p* < 0.05, compared with the control group.

**Figure 6 antioxidants-13-00599-f006:**
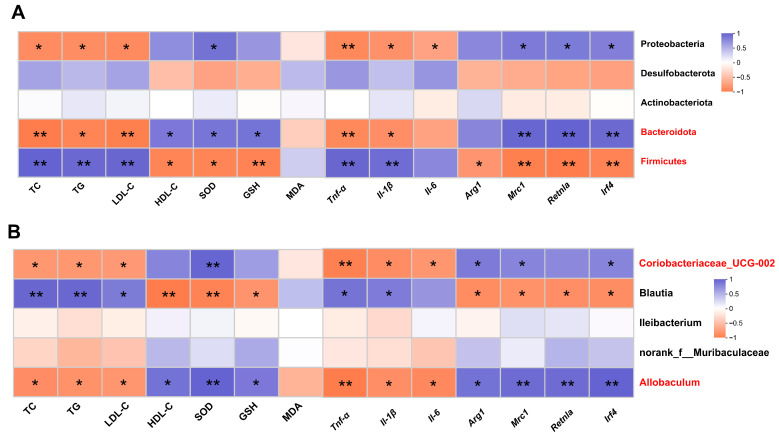
Correlation analysis between gut microbiota and metabolic phenotypes. (**A**) Spearman’s correlation analysis between five bacterial genera at the phylum level and metabolic phenotypes. (**B**) Spearman’s correlation analysis between five bacterial genera at the genus level and metabolic phenotypes. Blue revealed positive correlations, and orange revealed negative correlations. All data are presented as mean ± SEM. Statistical analysis was performed with a one-way ANOVA. * *p* < 0.05, ** *p* < 0.01, compared with the control group.

**Figure 7 antioxidants-13-00599-f007:**
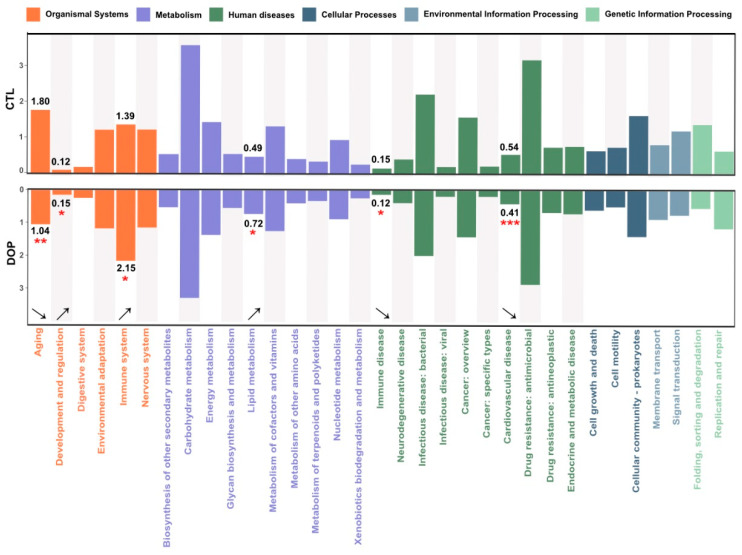
**Effects of DOP on microbial community functions are predicted by PICRUSt.** PICRUSt2 function prediction of the 16S rRNA gene from the gut microbiota. * *p* < 0.05, ** *p* < 0.01, *** *p* < 0.001, compared with the control group.

**Figure 8 antioxidants-13-00599-f008:**
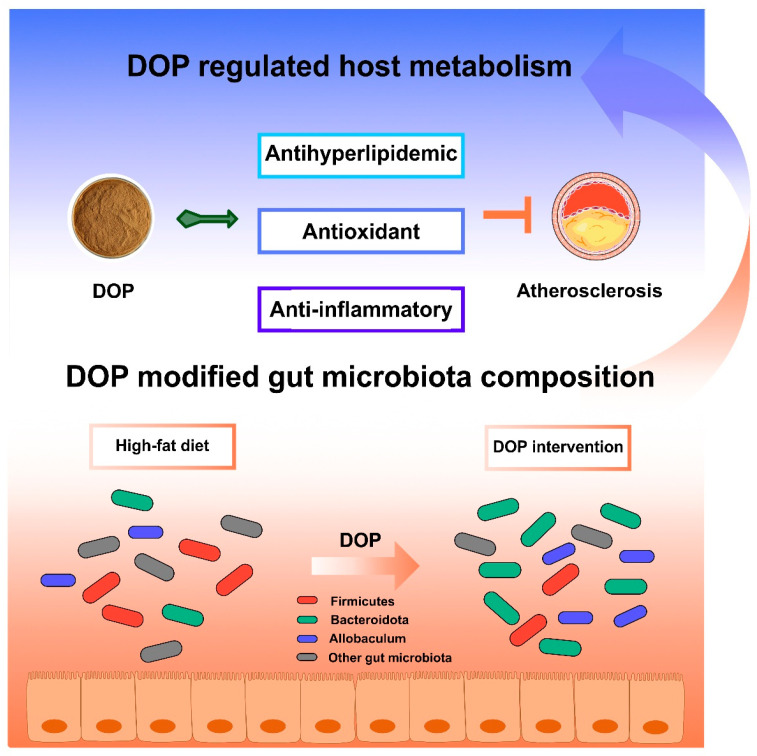
**Schematic illustration of DOP in ameliorating atherosclerosis.** The DOP treatment improved the serum lipid levels and antioxidants, increasing the anti-inflammatory factors, decreasing the inflammatory factors, and modifying the gut microbiota composition in atherosclerotic mice.

## Data Availability

Data are contained within the article and Appendix A.

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
