# Peer review of "The Antioxidant Dendrobium officinale Polysaccharide Modulates Host Metabolism and Gut Microbiota to Alleviate High-Fat Diet-Induced Atherosclerosis in ApoE−/− Mice"

_antioxidants, 2024, doi:10.3390/antiox13050599_

Round 1
Reviewer 1 Report
The manuscript by Qi et al. provides a comprehensive look at elements involved in atherosclerosis from genetic susceptibilty (ApoE-/- mice), dietary modulation with novel antioxidant, effects on oxidative stress levels, modulation inflammation/anti-inflammation processes, and effects on the microbiome. Including the microbiome is particularly relevant and an area of heightened research interest. The manuscript is well-organized and articulated. There are no major comments but rather numerous minor issues that should be easy to correct or consider as shown below.
The manuscript by Qi et al. entitled "The antioxidant Dendrobium officiale...," describes the pathological phenotype, serum lipid levels, antioxidant activities, anti-inflammatory levels, and fecal microbiota composition of atherosclerotic mice treated with saline alone or with DOP.
Is the Purpose on lines 20-21 intended to be a sentence fragment since others are not?
Could F/B on line 34 be written out at first mention or incorporated into sentences 36-37 as abbreviation following term?
Please review the use of a semi-colon versus comma throughout the abstract.
Please clarify sentence on line 62. Statins are a drug not blood lipid as implied by the current sentence construction.
Please consider other word choices for precious (line72) and valuable (line 73) to fully clarify what is meant. Also, look at valuable on line 75.
On line 82, "achieve the purpose" is not needed.
What is TFA (line 5)? What is LPMP methanol?
Line 114, mice were "gained?" Perhaps this should be obtained?
Please consider an alternative term for "executed" on line 129.
It would be helpful to include the wavelengths used in section 2.3 (line 140).
Regarding the scientific merit of the figures and tables, thee are no major comments. However, there is a need for careful proofreading for spelling and format.
Author Response
We would like to sincerely thank all the reviewers for their constructive comments and insightful suggestions that allowed us to substantially improve our manuscript. In light of these comments, we have conducted additional experiments and introduced textual changes to the manuscript to improve its preciseness and clarity. All the figure numbers are those of the revised manuscript. Revisions and changes to the manuscript are clearly highlighted. The following is our point-by-point response.
Response to Reviewer #1:
- Major comments
- Is the Purpose on lines 20-21 intended to be a sentence fragment since others are not?
Response: Thank you for this comment. As suggested, we have rewritten the Purpose in the Abstract. The following are the modified sentences: “This study aims to investigate the inhibitory effect and the potential mechanism of DOP on the high-fat-diet-induced atherosclerosis in Apolipoprotein E knockout (ApoE–/–) mice.” (page 1 of the revised manuscript)
- Could F/B on line 34 be written out at first mention or incorporated into sentences 36-37 as abbreviation following term?
Response: Thank you for this suggestion. The following are the modified sentences: “we found that DOP restructures the gut microbiota composition by decreasing the Firmicutes/Bacteroidota (F/B) ratio.” (page 1 of the revised manuscript).
- Please review the use of a semi-colon versus comma throughout the abstract.
Response: Thank you for this suggestion. We have changed the semi-colon into comma throughout the abstract (page 1 of the revised manuscript).
- Please clarify sentence on line 62. Statins are a drug not blood lipid as implied by the current sentence construction.
Response: Thank you for this suggestion. We agree and have rewritten this sentence as suggested. The following are the modified sentences: “The treatment of atherosclerosis is mainly based on etiological treatment. For example, statins intervene with atherosclerosis by lowering blood lipid levels.” (page 2 of the revised manuscript).
- Please consider other word choices for precious (line72) and valuable (line 73) to fully clarify what is meant. Also, look at valuable on line 75.
Response: Thank you for this suggestion. We have rewritten this paragraph as suggested. The following are the modified paragraph: “Dendrobium officinale has been widely studied for its commercial and medicinal value. Dendrobium officinale is a folk herbal medicine and contains many anti-oxidative compounds, such as alkaloids, flavonoids, and polysaccharides.” (page 2-3 of the revised manuscript).
- On line 82, "achieve the purpose" is not needed.
Response: Thank you for this suggestion. We have rewritten this paragraph as suggested. The following are the modified paragraph: “The non-digestible components and fermentation products of polysaccharides can further regulate the composition of intestinal flora, improve intestinal function, and maintain body health.” (page 3 of the revised manuscript).
- What is TFA (line 5)? What is LPMP methanol?
Response: The full name of TFA is trifluoroacetic acid. The full name of PMP is 1-phenyl-3-methyl-5-pyrazolone. We have rewritten this part in the “Materials and methods” section as suggested. The following are the modified paragraph: “In detail, 3 ml of trifluoroacetic acid (TFA, 2 M) was added to 2 mg of DOP and hydrolyzed in a sealed tube at 120℃ for 3 h. After cooling, the sample was mixed with methanol, and the evaporation was repeated three times to remove excess TFA. Added 250 μL of 0.6 mol/L NaOH and 500 μL of 0.4 mol/L 1-phenyl-3-methyl-5-pyrazolone (PMP) methanol solution to the hydrolysate or standard monosaccharide mixture (1 mg/ml).” (page 3 of the revised manuscript).
- Line 114, mice were "gained?" Perhaps this should be obtained?
Response: Thank you for this suggestion. We have replaced “gained” with “obtained” as suggested (page 3 of the revised manuscript).
- Please consider an alternative term for "executed" on line 129.
Response: Thank you for this suggestion. We have rewritten this sentence as suggested. The following are the modified paragraph: “After anesthesia, the mice were sacrificed.” (page 4 of the revised manuscript).
- It would be helpful to include the wavelengths used in section 2.3 (line 140).
Response: We agree and have added the wavelengths used in section 2.3. (page 4 of the revised manuscript).
- Detail comments
- Regarding the scientific merit of the figures and tables, there are no major comments. However, there is a need for careful proofreading for spelling and format.
Response: Thank you for your kind reminder. We have carefully checked all the spelling and format in the revised manuscript.
Reviewer 2 Report
Dear authors, two control groups are missing: Animals without a High Fat diet, supplemented or not with the extracts.
Furthermore, biochemical parameters (glucose levels for example) of the animlas are not presented.
Microbiota data changes are small and they are partially in disagreement with the micribiota date presented by the same authors in a very related manuscript (https://doi.org/10.26599/FSHW.2024.9250007).
Data regarding biochemical parameters of the experimental groups, for example glucose values are missing.
Also, the data presented from microscopy images would need to be also supported by more quantitative techniques.
Author Response
We would like to sincerely thank all the reviewers for their constructive comments and insightful suggestions that allowed us to substantially improve our manuscript. In light of these comments, we have conducted additional experiments and introduced textual changes to the manuscript to improve its preciseness and clarity. All the figure numbers are those of the revised manuscript. Revisions and changes to the manuscript are clearly highlighted. The following is our point-by-point response.
Response to Reviewer #2:
- Major comments
- Dear authors, two control groups are missing: Animals without a High Fat diet, supplemented or not with the extracts.
Response: We sincerely thank you for this constructive comment. We did not include the two control groups due to the following two reasons. Firstly, the purpose of this study is to investigate the effect of DOP on the established atherosclerosis. To this end, we induced a mouse model of atherosclerosis in ApoE–/– mice with HFD for 10 weeks, and subsequent DOP intervention for an additional 8 weeks. Therefore, the entire experiment lasts for 18 weeks. However, 18 weeks of normal diet is not enough to induce an obvious fibrous plaque in ApoE-/-mice(https://doi.org/10.1161/01.atv.14.1.133; https://doi.org/10.1161/01.atv.18.5.842). Secondly, in terms of side effects, 200 mg/kg DOP used in this study, even a higher dose (400 mg/kg) used in other studies, showed no obvious side effects in mice according to previous reports(https://doi.org/10.1016/j.fochx.2022.100235R;https://doi.org/10.1016/j.ijbiomac.2023.126920; https://doi.org/10.26599/FSHW.2024.9250007). Therefore, the two control groups were not used in this study. We added additional text and cited relevant references to make it clear (page 6 of the revised manuscript).
- Furthermore, biochemical parameters (glucose levels for example) of the animals are not presented.
Response: Thanks for your suggestion. We have performed additional experiments on biochemical parameters (glucose levels and glycosylated serum protein) as suggested. The new results have been included in the new version of the revised manuscript (page 7 of the revised manuscript and Supplemental Figure 3).
- Microbiota data changes are small and they are partially in disagreement with the microbiota data presented by the same authors in a very related manuscript (https://doi.org/10.26599/FSHW.2024.9250007).
Response: Thanks for your suggestion. However, the author Xiaoxia Chen (Guangdong Province Key Laboratory for Green Processing of Natural Products and Product Safety, Guangzhou 510640, China) in the article you mentioned is not the same person as our corresponding author Xiaoxia Chen (Department of Radiology, the Third Medical Centre, Chinese PLA General Hospital, Beijing 100039, China). In addition, the reasons for the different Microbiota data may be due to the following aspects, including DOP purity and composition, mouse genetic background, disease models used, and DOP dosages:
|
factors |
https://doi.org/10.26599/FSHW.2024.9250007 |
This study |
|
DOP |
Purity, monosaccharide composition and molecular weight are different |
|
|
Mice |
C57BL/6J mice |
APOE-/- mice |
|
Disease model |
Type 2 diabetes mice model |
Atherosclerosis mice model |
|
DOP dosage |
400 mg/kg |
200 mg/kg |
- Detail comments
- Data regarding biochemical parameters of the experimental groups, for example glucose values are missing.
Response: Thanks for your suggestion. As suggested, we performed additional experiment and added the biochemical parameters of mice (glucose levels and glycosylated serum protein) into the revised manuscript (page 7 of the revised manuscript and Supplemental Figure 3).
- Also, the data presented from microscopy images would need to be also supported by more quantitative techniques.
Response: Thanks for your kind suggestion. As for the quantification of microscopy images, we additionally quantified fluorescence intensity of mac-3 and α-SMA staining to further confirm the reduced macrophage content and thicker fibrous cap upon DOP treatment. The new data have been added into the revised Figure 4 (page 10 of the revised manuscript).
Reviewer 3 Report
The manuscript by Qi et al. " The antioxidant polysaccharide Dendrobium officinale (DOP) modulates host metabolism and gut microbiota to attenuate high-fat diet-induced atherosclerosis in ApoE-/- mice". Dop (200 mg/kg by gavage) was tested to inhibit atherosclerosis in mice. The anti-atherosclerotic effect of DOP in vivo in 2 groups of ApoE-/- mice (without DOP and +DOP after a high-fat diet. The atherosclerotic effects of a high-fat diet and that of DOP were compared using histological methods (H&E and masson staning) to characterize the lesion area and plaque stability in blood vessels respectively. The mechanism of action of DOP was studied at the level of inflammatory and anti-inflammatory factors by immunofluorescence with 7 specific antibodies, and the antioxidant effect at the level of a biomarker (MDA levels) and the activities of 2 enzymes GSH-PX and SOD). In addition, the fecal microbiota was compared between the 2 experimental groups using 16S rRNA gene sequencing.
Results obtained after 8 weeks confirm a positive effect of DOP on serum levels of TC, TG, LDL-C and HDL-C. All comparisons presented in this article are statistically significant.
The article is very clear, well written and attractively presented with an appropriate design.
General comments: DOP has been characterized for its monosaccharide composition. This implies that the mechanism of action is primarily a prebiotic effect. This hypothesis is not presented and should be discussed.
Sperman's correlation analyses between 5 bacterial genera (fig 6) are multiple. These correlations should be presented as possible avenues for more direct analysis of these associations. This limitation should be discussed.
Is DOP composed exclusively of monosacharides and not polysaccharides?
Only one dose of DOP is used here. Please explain this choice and is there a known dose-response relationship, what toxicity is known for this compound?
Other constituents in other studies and characterization of DOP should be discussed as well as their physiological benefits.
Minor points
Line 223: considerably is excessive.
Line 114, the choice of ApoE-/- mice should be better explained.
Author Response
Response to Reviewer #3:
- General comments:
- DOP has been characterized for its monosaccharide composition. This implies that the mechanism of action is primarily a prebiotic effect. This hypothesis is not presented and should be discussed.
Response: Thanks for your suggestion. We agree and have added additional text into the Discussion section. The following are the modified paragraph: “Previous study has demonstrated that DOP was indigestible and non-absorbing after oral administration (doi:10.1021/acs.jafc.9b01489). However, intestinal flora could degrade the DOP into short-chain fatty acids (SCFAs), exhibiting the potential of prebiotics to maintain intestinal homeostasis (doi:10.1016/j.jep.2018.01.021) (see page 17 of the revised manuscript).
- Sperman's correlation analyses between 5 bacterial genera (fig 6) are multiple. These correlations should be presented as possible avenues for more direct analysis of these associations. This limitation should be discussed.
Response:We agree with your suggestion and have added this part in the discussion section (page 17 of the revised manuscript).
- Is DOP composed exclusively of monosacharides and not polysaccharides?
Response: Thanks for this comment. Actually, DOP is not a single polysaccharide composed exclusively of monosaccharides. It is a mixture of polysaccharides that have different degrees of polymerization (doi: 10.1016/j.ijbiomac.2016.06.100, doi: 10.3389/fnut.2022.965073, doi: 10.1186/s40538-02100214- x).
- Only one dose of DOP is used here. Please explain this choice and is there a known dose-response relationship, what toxicity is known for this compound?
Response: Thanks for this helpful comment. The dosage design in this study was mainly based on the published paper (doi: 10.1016/j.ijbiomac.2023.125787). They calculated the safe starting dose in human clinical trials by using the total polysaccharide content of DOP according to the published experimental studies. Animal drug doses were then estimated by scaling factor (human clinical dose: animal dose = 1:33), converting human clinical doses to animal doses (doi:10.1002/ddr.21461). Therefore, the appropriate DOP dose was determined to be 200 mg/kg. In addition, it was reported that DOP could dose-dependently reduce inflammation (https://doi.org/10.1016/j.carbpol.2018.01.013) and oxidative stress levels (https://doi.org/10.1016/j.cbi.2021.109615). Among them, DOP had a significant inhibitory effect when the dose reached 200 mg/kg. In addition, no obvious toxicity of DOP in mice was observed when the dosage of DOP reached 400 mg/kg (https://doi.org/10.1016/j.ijbiomac.2023.126920,https://doi.org/10.26599/FSHW.2024.9250007). We have added additional text into the Methods section to make it clear (page 4 of the revised manuscript).
- Other constituents in other studies and characterization of DOP should be discussed as well as their physiological benefits.
Response: Thanks for this constructive comment. We have added additional text into the discussion section regarding these points (page 17 of the revised manuscript).
- Minor points
- Line 223: considerably is excessive.
Response: We deleted “considerably” as suggested (page 7 of the revised manuscript).
- Line 114, the choice of ApoE-/- mice should be better explained.
Response: We have explained the choice of ApoE-/- mice in the results section and inserted two relevant references. The following are the modified paragraph: “ApoE–/– mice have been frequently used in atherosclerosis research (https://doi.org/10.1016/j.ejphar.2017.05.010; https://doi.org/10.1161/ATVBAHA.111.237693)” (page 7 of the revised manuscript).
Reviewer 4 Report
The current experimental study in animal models is intriguing. Dendrobium officinale is a precious traditional Chinese medicine. The principal active components are polysaccharides (DOP), which are highly potent in therapeutic applications. A growing number of studies have revealed the potential health benefits of DOP, which is reported to have diverse biological activities, including anticancer, anti-inflammatory, antioxidant, and immunomodulatory effects.
My comments are:
How did the researchers define the purity of DOP?
The authors should describe the precise mechanism of how these orally ingested polysaccharides work in vivo. Could they be absorbed alone or with food?
The authors should report the blood glucose values from the experimental animals. The literature shows that DOP has hypoglycemic activity, so the reported improvements could be due to lower blood glucose measurements.
The manuscript is well written, and the discussion/conclusions are acceptable.
Overall, the data are of interest for future research.
The current experimental study in animal models is intriguing. Dendrobium officinale is a precious traditional Chinese medicine. The principal active components are polysaccharides (DOP), which are highly potent in therapeutic applications. A growing number of studies have revealed the potential health benefits of DOP, which is reported to have diverse biological activities, including anticancer, anti-inflammatory, antioxidant, and immunomodulatory effects.
My comments are:
How did the researchers define the purity of DOP?
The authors should describe the precise mechanism of how these orally ingested polysaccharides work in vivo. Could they be absorbed alone or with food?
The authors should report the blood glucose values from the experimental animals. The literature shows that DOP has hypoglycemic activity, so the reported improvements could be due to lower blood glucose measurements.
The manuscript is well written, and the discussion/conclusions are acceptable.
Overall, the data are of interest for future research.
Author Response
Response to Reviewer #4:
- How did the researchers define the purity of DOP?
Response: Thank you for this helpful comment. The purity of polysaccharides was determined using the phenol‑sulfuric acid method. We have inserted a reference to make it clear. Reference: Masuko T, Minami A, Iwasaki N, Majima T, Nishimura S, Lee YC. Carbohydrate analysis by a phenol-sulfuric acid method in microplate format. Anal Biochem. 2005;339(1):69-72. doi:10.1016/j.ab.2004.12.001.
- The authors should describe the precise mechanism of how these orally ingested polysaccharides work in vivo. Could they be absorbed alone or with food?
Response: Thank you for this helpful comment. We have added additional text into the Discussion section (page 17 of the revised manuscript). The following are the added paragraph: Previous study has demonstrated that DOP was indigestible and non-absorbing after oral administration (doi:10.1021/acs.jafc.9b01489). However, intestinal flora could degrade the DOP into short-chain fatty acids (SCFAs), exhibiting the potential of prebiotics to maintain intestinal homeostasis (doi:10.1016/j.jep.2018.01).
- The authors should report the blood glucose values from the experimental animals. The literature shows that DOP has hypoglycemic activity, so the reported improvements could be due to lower blood glucose measurements.
Response: Thanks for your suggestion. We performed additional experiment and added the glucose and glycosylated serum protein levels into the revised manuscript (page 7 of the revised manuscript and Supplemental Figure 3). The results showed that the oral administration of DOP had no significant hypoglycemic effect (P > 0.05). However, previous study has reported that DOP exerts hypoglycemic effects on type 2 diabetes mellitus (T2DM) mice(https://doi.org/10.26599/FSHW.2024.9250007). There may be several reasons for the differences in results, including DOP purity and composition, mouse genetic background, disease models used, and DOP dosages:
|
factors |
https://doi.org/10.26599/FSHW.2024.9250007 |
Our article |
|
DOP |
Purity, monosaccharide composition and molecular weight are different |
|
|
Mice |
C57BL/6J mice |
APOE-/- mice |
|
Disease model |
type 2 diabetes mice model |
Atherosclerosis mice model |
|
DOP dosage |
400 mg/kg |
200 mg/kg |
Round 2
Reviewer 2 Report
Dear Authors:
Still, a control group is missing. This group without an HFD treatment is needed to evaluate the relevance of the changes that the treatment produces in the animals. The authors indicate that the treatment reduces atherosclerosis, but up to what extent compared with control group?
Again biochemical parameters of the animals are missing. For example, Glucose values are indicated as percentages but no real values are presented. Furthermore, other parameters such as cholesterol, TG, fatty acid composition, GLP1, or short-chain fatty acids in feces would be relevant to understanding the role of microbiota in the response to treatment. An HFD should induce insulin resistance (at some point this model is partially similar to others used previously for type 2 diabetes) and therefore, the effects of the extracts on all these parameters are important.
Regarding the effects on atheroscleroses, not only a quantitation of the images but the measurement of lipid levels in the vasculature by biochemical methods would have strenght the manuscript.
No minor points are detected.
Reviewer 3 Report
This study is of interest for many researchers. To my knowledge, the sugars present in the tea is now stu studied and present interest for the microbiote modulation. This is a preclinical study before going to clinical trials. The experimental model is a good choice.
Nothing to comment as my comments have been taken into consideration.
Author Response
We sincerely thank you for your constructive comments, which have enabled us to greatly improve our manuscript.
Reviewer 4 Report
no other comments
no other comments
Author Response
Thanks for your constructive comments and insightful suggestions again.
Round 3
Reviewer 2 Report
New version of the manuscript is improved. Albeit there are some points that could be addreseed in more detail (inclusion of a HF control group, more biochemical parameters and in situ assays). The results provides enough evidence to support the hypothesis.
No detailed comments